

# Inter-subspecies diversity of maize to drought stress with physio-biochemical, enzymatic and molecular responses

Gokhan Eskikoy and Imren Kutlu

Field Crops Department/Faculty of Agriculture, Osmangazi University, Eskişehir, Turkey

Corresponding author
Imren Kutlu, ikutlu@ogu.edu.tr

## ABSTRACT

**Background:** Drought is the most significant factor limiting maize production, given that maize is a crop with a high water demand. Therefore, studies investigating the mechanisms underlying the drought tolerance of maize are of great importance. There are no studies comparing drought tolerance among economically important subspecies of maize. This study aimed to reveal the differences between the physio-biochemical, enzymatic, and molecular mechanisms of drought tolerance in dent (*Zea mays indentata*), popcorn (*Zea mays everta*), and sugar (*Zea mays saccharata*) maize under control (no-stress), moderate, and severe drought stress.

**Methods:** Three distinct irrigation regimes were employed to assess the impact of varying levels of drought stress on maize plants at the V14 growth stage. These included normal irrigation (80% field capacity), moderate drought (50% field capacity), and severe drought (30% field capacity). All plants were grown under controlled conditions. The following parameters were analyzed: leaf relative water content (RWC), loss of turgidity (LOT), proline (PRO) and soluble protein (SPR) contents, membrane durability index (MDI), malondialdehyde (MDA), and hydrogen peroxide ($H_2O_2$) content, the antioxidant enzyme activities of superoxide dismutase (SOD), ascorbate peroxidase (APX), and catalase (CAT). Additionally, the expression of heat shock proteins (HSPs) was examined at the transcriptional and translational levels.

**Results:** The effects of severe drought were more pronounced in sugar maize, which had a relatively high loss of RWC and turgor, membrane damage, enzyme activities, and HSP90 gene expression. Dent maize, which is capable of maintaining its RWC and turgor in both moderate and severe droughts, and employs its defense mechanism effectively by maintaining antioxidant enzyme activities at a certain level despite less MDA and $H_2O_2$ accumulation, exhibited relatively high drought tolerance. Despite the high levels of MDA and $H_2O_2$ in popcorn maize, the up-regulation of antioxidant enzyme activities and HSP70 gene and protein expression indicated that the drought coping mechanism is activated. In particular, the positive correlation of HSP70 with PRO and HSP90 with enzyme activities is a significant result for studies examining the relationships between HSPs and other stress response systems. The discrepancies between the transcriptional and translational findings provide an opportunity for more comprehensive investigations into the role of HSPs in stress conditions.

## INTRODUCTION

Maize (*Zea mays* L.) is currently the most produced cereal, with three (dent maize (*Zea mays indentata*), popcorn maize (*Zea mays everta*), and sugar maize (*Zea mays saccharata*)) of its seven botanical subspecies being of high economic importance (*Pandita et al., 2023*). Maize has long been of significant cultural and economic importance, serving as a food source for humans and animals, as well as a key ingredient in numerous industrial products. It is also cultivated on a vast scale. However, climate change causes abiotic issues such as drought stress, which poses a significant risk to maize production (*Kim & Lee, 2023*).

Approximately 15% of global maize yield losses are attributed to drought stress. The spread of numerous diseases in hot climates and the drying up and warming of maize production areas will greatly impact maize yield. To maintain maize yield in dry areas and stabilize predicted yield losses, it is necessary to enhance the drought resistance of maize (*Adewale et al., 2018*; *Rasheed et al., 2023*). Drought tolerance is the result of a plant's attempts to survive or recover from stress and allows plants to grow and maintain relatively high yields even in the face of drought. A plant is considered drought-acclimated if the tolerance is limited to that specific generation. A plant genotype is considered to be adapted to drought circumstances if it continues across several generations (*Seleiman et al., 2021*). In response to drought stress, plants attempt to maintain their cell water status and turgor by regulating osmotic pressure as their initial response (*Sanders & Arndt, 2012*). In order to accurately assess osmotic adjustment, it is essential to evaluate several key parameters, including leaf relative water content (RWC), loss of turgidity (LOT), and the presence of osmotic regulatory solutes such as proline (PRO). Other stress response mechanisms include membrane damage (MD), lipid peroxidation, accumulation of reactive oxygen species (ROS), activation of antioxidants, and the expression of certain genes associated with drought stress.

The RWC is a more consistent indicator across organs, populations, and potentially even species than other less integrative indicators. This is because it captures the combined impact of the physiological and morphological changes that plants make to preserve water balance and avoid turgor loss (*Sapes & Sala, 2021*). When the RWC reaches the point of LOT, plants are likely to exhibit an elevated risk of mortality due to the necessity of maintaining cell turgor through the maintenance of cell water volume above specific thresholds (*Lambers & Oliveira, 2019*). Plants utilize the accumulation of PRO to facilitate osmotic adjustment as a means of defense against drought stress. When a cell experiences osmotic stress, PRO is transported into the cytoplasm, and the cytoplasmic concentration is increased, thereby lowering the osmotic potential. This allows the cell to continue absorbing extracellular water even when its osmotic potential is low, maintaining the osmotic balance between the cell's protoplasm and the surrounding environment (*Shakeri et al., 2019*). The presence of water molecules on the surface of proteins can facilitate the

formation of a protective layer that can bind to PRO. This process serves to decrease water loss and restrict the flow of water to the outside of the cell, thereby creating a protective membrane. Furthermore, the protective membrane preserves the high structure and activity of biological macromolecules by providing effective protection for proteins and other macromolecules. Proline can combine with denatured proteins to increase their hydrophilicity when the proteins are under stress from adversity (*Yang et al., 2021*).

Under drought stress, plant cell membranes become more susceptible to rapid damage and leakage. This leakage in the membrane is caused by an uncontrolled increase of free radicals, which cause lipid peroxidation. Damage to the fatty acids of the cell membrane can result in the formation of small hydrocarbons, such as malondialdehyde (MDA). MDA is the final product of lipid peroxidation and serves as a marker of membrane cellular damage (*Gharibi et al., 2016*). It is well established that the production of hydrogen peroxide ($H_2O_2$) and other toxic oxygen species, which cause lipid peroxidation and oxidative damage, is also triggered by drought stress. Given that $H_2O_2$ is a powerful oxidant, it can initiate localized oxidative damage in leaf cells, resulting in impairment of metabolic function and loss of cellular integrity (*Černý et al., 2018*).

Water stress results in a cellular redox imbalance due to an increase in the generation and accumulation of ROS, which disrupts metabolic functions and damages proteins, RNA, DNA, and cellular membranes (*Choudhury et al., 2017*). However, ROS damage can be avoided or attenuated by activating both enzymatic and non-enzymatic antioxidant defense systems. Superoxide dismutase (SOD), catalase (CAT), and ascorbate peroxidase (APX) enzymes are only a few of the unique ROS elimination systems found in plants that are essential for scavenging ROS. These enzymes can be detected extracellularly in the apoplast and at the plasma membrane, in addition to being present in several plant cell compartments (*Hasanuzzaman et al., 2020*). The initial line of defense against ROS is SOD, which functions by accelerating the conversion of superoxide radicals ($O_2^{\cdot-}$) into $H_2O_2$ and molecular oxygen ($O_2$). Since $H_2O_2$ is also a reactive species, CAT and APX transform it into $O_2$ and $H_2O$ (*Rajput et al., 2021*). SOD and CAT are present in several subcellular sites because they function in concert as the first line of antioxidant defenses. Drought tolerance has been linked to an efficient ROS scavenging ability that lessens the detrimental effects of such molecules. Moreover, alterations in the metabolism of oxidants and antioxidants brought on by abiotic stressors might operate as biochemical stress markers in plants (*de Araújo Silva et al., 2021*).

Significant alterations in the expression of stress-related genes are typically the result of molecular reactions that arise during drought stress (*Singh et al., 2019*). Heat shock proteins (HSPs) are the most frequently occurring molecular components among stress-related genes that exhibit upregulation in response to nearly all forms of stress experienced by plants (*Priya et al., 2019*). The multigene families HSP70 and HSP90 are highly conserved and comprise both constitutive and inducible members. These groups of proteins serve as molecular chaperones in the endoplasmic reticulum, plastids, mitochondria, and cytoplasm/nuclear membrane. The aforementioned proteins facilitate intracellular translocation, signal transduction, and protein folding under typical growth circumstances (*Kozeko, 2021*). It is well established that a group of signaling and regulatory

proteins serve as HSP90 substrates. In unfavorable circumstances, inducible chaperones are added to the cellular pool, where they play a crucial role in halting the aggregation of damaged proteins and promoting their refolding. In addition, HSPs are involved in a variety of functions, most notably stomatal closure regulation and autoregulation of the heat shock response (*Augustine, 2016*). An increasing body of research has demonstrated that plants responses to water deficits involve HSPs. Several plant species have been shown to up-regulate specific HSP70s when dehydrated (*Augustine et al., 2015*; *Chaudhary et al., 2019*; *Landi et al., 2019*). In particular, fewer studies have demonstrated up-regulation of certain HSP90s during water deficit in Arabidopsis (*Swindell, Huebner & Weber, 2007*), rice (*Hu, Hu & Han, 2009*; *Zou et al., 2009*), potatoes (*Ambrosone et al., 2013*), and barley (*Chaudhary et al., 2019*). *Li & Howell (2021)* reported that the expression of HSP90, HSP70, and small HSPs was induced in order to prevent cellular damage in the drought-sensitive genotype of maize. In contrast, the expression of small HSPs was found to be up-regulated in the drought-tolerant genotype under drought stress.

Dent maize, popcorn maize, and sugar maize subspecies differ in their plant characteristics, growth and development, tolerance to stress conditions, as well as intended use. Previous studies have only compared different varieties of a single subspecies in terms of drought tolerance (*Anjum et al., 2016*; *Kamphorst et al., 2019*; *Ali et al., 2023*; *Mousavi et al., 2023*; *Schmitt, do Amaral Junior & Kamphorst, 2024*). There is currently no study comparing the different subspecies to each other under drought stress. In addition, the role of heat-inducible HSPs in the protection of cells under dehydration remains poorly understood. Further investigation is required to elucidate the interplay between HSPs and other stress response mechanisms in order to ascertain the mechanisms underlying drought stress tolerance in maize. The objective of this study was to characterize different maize subspecies in terms of several physio-biochemical parameters, including leaf water status, PRO content, cell damage, and antioxidant enzyme activities. Additionally, expression changes in HSPs were evaluated by quantitative reverse transcription polymerase chain reaction (RT-qPCR) and western blot (WB) analyses. This result may enhance our comprehension of the molecular basis of drought tolerance and could be utilized for a prospective breeding program in maize.

# MATERIALS AND METHODS

## Conducting the experiment

The soil utilized in this study was obtained from the experimental research area of Eskişehir Osmangazi University at a depth of 0–30 cm. It was prepared by sieving through a 2 mm filter after allowing it to dry in the open for a week. The soil is free of salt and has a sandy loam texture. The pH of the soil is 7.99, indicating an alkaline nature. Its organic matter content is low, at 1.08%. Additionally, the soil is calcareous, with a calcium content of 1.98%. It contains adequate levels of potassium, copper, and iron, but not enough phosphorus, manganese, or zinc.

One kilogram of soil was saturated with water and placed in a container with perforations at the base to ascertain the field capacity of the soil. After a 24-h period, the water that gravitationally drained was collected, and the field capacity of the soil was

determined using the difference method, as outlined in the following formula (*Danish et al., 2020*).

$$Field\ capacity\ (\%) = \frac{Water\ added\ (ml) - Water\ infiltrated\ in\ 24\ hours\ (ml)}{Soil\ weight\ (g)} \times 100$$

The study employed three distinct subspecies of maize: dent maize (*Zea mays indendata*, cv. P0937), popcorn maize (*Zea mays everta*, cv. Baharcin), and sugar maize (*Zea mays saccharata* Sturt, cv. Adapare). The maize varieties utilised in the study are commonly cultivated in Türkiye and are known as drought tolerant.

In order to investigate the drought tolerance of the intersubspecies, the ones of every subspecies that were drought-tolerant were chosen.

Ten-liter plastic containers were utilized to cultivate maize seedlings, containing a mixture of 2:1:1:1 soil, sand, peat, and perlite. A basic solution containing macro- and micro-elements was used for fertilization of the pots. The solution included 200 mg of nitrogen $(NH_4)_2SO_4$, 125 mg of potassium, 100 mg of phosphorus $(KH_2PO_4)$ per kg of soil, 2.5 mg of iron (Fe-EDTA), and 5 mg of zinc $(ZnSO_4.7H_2O)$ per kg of soil. A total of 12 maize seeds were planted in each pot for each treatment and replication. Following the emergence of the plants, the number of plants per pot was reduced to six. The plants were maintained at a temperature of 25/17 °C, with a light intensity of 800 µmol $m^{-2}$ $s^{-1}$, a humidity of 65%, and a light/dark cycle of 14/10 h throughout the duration of the experiment. The Hoagland solution was applied once a week until harvest, comprising 5 mM $KNO_3$, 1 mM $KH_2PO_4$, 5 mM $Ca(NO_3)_2$, 2 mM $MgSO_4$, 50 µM $H_3BO_3$, 10 µM $MnCl_2$, 1 µM $ZnSO_4$, 0.4 µM $CuSO_4$, 0.1 µM $H_2MoO_4$, and 20 µM Fe-EDTA.

The maize plants were grown under standard irrigation conditions, which included maintaining 80% field capacity for the initial 14 days following germination and emergence until the V3 stage. Throughout the drought trial, the moderate drought group was maintained at 50% field capacity, the severe drought group at 30% field capacity, and the control group was maintained at 80% field capacity. From the time of plant emergence until the V14 stage, which denotes the end of the vegetative cycle, the plants were subjected to 50 days of drought treatments. Namely, the maize plants were exposed to drought stress from V3 to V14. At the conclusion of the designated period, the plants were harvested for subsequent analysis.

## Osmoregulation parameters

In order to determine the RWC and LOT values of the plants, three discs with a diameter of 2 cm were removed from the leaf samples. The fresh weights (FW) immediately, the turgor weights (TW) after being kept in pure water for 4 h, and the dry weights (DW) after being kept at 70 °C for 24 h were recorded. The data obtained were used to calculate the RWC and LOT values using the following formulas, with the resulting values expressed as a percentage (*Gulen & Eris, 2003*).

$$RWC = (FW - DW)/(TW - DW) \times 100$$

$$LOT = (TW - FW)/TW \times 100$$

Proline analysis was conducted in accordance with the methodology described by *Bates, Waldren & Teare (1973)*. For the analysis, 0.2 g of plant leaves were homogenized with 4 mL of 3% sulfosalicylic acid and centrifuged (Neofuge-13R) at 6,000 rpm for 10 min. Two milliliters of the supernatant were transferred to glass test tubes and 2 mL of glacial acetic acid and 2 mL of acid ninhydrin were added. The tubes were covered with aluminum foil and placed in a water bath at 100 °C for 1 h. The samples were then cooled in an ice bath for 10 min. Four milliliters of toluene were added to the cooled samples, which were then shaken gently and left for 5 min. Once two phases were observed in the tubes, samples were taken from the upper phase and concentrations were determined at 520 nm with a spectrophotometer (Thermo-Aquamate) on the calibration curve prepared previously using PRO standards. The amount of PRO was calculated using the formula: [(μg proline/ ml × ml toluene)/115.5 μg/μmol]/[(g sample)/5] = μmol proline/g fresh weight.

The soluble protein (SPR) concentrations of leaf samples from all treatments were determined using a bovine serum albumin (BSA) standard according to the method of *Bradford (1976)*. The absorbance values of the solution were read at a wavelength of 595 nm in a spectrophotometer (Thermo-Aquamate). The amount of SPR was calculated against a standard curve generated using BSA standards.

## Membrane injury parameters

The MDI in the leaves of the plants was determined according to the methodology described by *Sairam, Shukla & Saxena (1997)*. For this purpose, the leaves were harvested at random from the plants in each condition and carefully cut with a scalpel. The leaf samples were then washed thoroughly with pure water at least six to seven times. This procedure ensured that cellular fluid contamination from the cut portions of the leaves did not affect the MDI values. Subsequently, 2 cm leaf sections were carefully excised from the aforementioned samples, placed in glass tubes, and 20 ml of pure water was added. The samples were then shaken on a shaker for 4 h, after which the relative ion amount (A) passing into pure water was measured with an electrical conductivimeter (S230-K; Mettler-Toledo). Then, the identical samples were maintained in a 100 °C water bath (Nuve, Turkey) for 30 min, then allowed to cool to room temperature before being subjected to a second measurement of the relative ion content (B). Finally, the MDI of the leaves was calculated using the following formula:

$$MDI = (1 - A/B) \times 100$$

The extent of lipid peroxidation in the leaf tissues of the plants in the control and stress groups will be quantified by measuring the amount of MDA using the method described by *Hodges et al. (1999)*. The method entails the homogenization of 0.1 g of fresh leaf samples taken from the leaves of the control and stress groups in three replicates with 1.5 ml of 5% trichloroacetic acid (TCA) on ice after being cut into small pieces. The aforementioned mixture was then centrifuged at 12,000 rpm at 25 °C for 15 min, after which the supernatant was utilized to ascertain the quantity of MDA. Equal volumes of the supernatant and a 20% TCA solution containing 0.5% thiobarbituric acid (TBA) were transferred to new tubes. Subsequently, the tubes were placed in a water bath at 95 °C for

30 min, after which they were transferred to an ice bath to halt the reactions within the tubes. Subsequently, the tubes were centrifuged at 1,000 rpm for 5 min, and the absorbance was measured at 532 and 600 nm wavelengths in a spectrophotometer. A 20% TCA solution containing 0.5% TBA will be employed as a blind. The MDA content (nmol gFW$^{-1}$) in leaf tissues will be calculated according to the following formula:

$$\text{MDA content} = [(A532 - A600) \times \text{extraction volume}]/[155 \times \text{sample quantity}]$$

The $H_2O_2$ levels were determined in accordance with the methodology described by *Velikova, Yordanov & Edreva (2000)*. For this purpose, leaf tissues (500 mg) were homogenized with 5 mL of 0.1% TCA. The homogenate was then centrifuged at 12,000 g for 15 min. A volume of 0.5 mL of the supernatant was combined with 0.5 mL of a 10 mM potassium phosphate buffer solution (pH 7.0) and 1 mL of 1 M KI. The absorbance of the supernatant was measured using a spectrophotometer with a wavelength of 390 nm. The quantity of $H_2O_2$ was determined by means of a standard curve.

## Antioxidant enzyme activities

The antioxidant enzyme activities (SOD, APX, and CAT) were analyzed according to the method described by *Cakmak & Marschner (1992)*.

For the extraction of the enzymes, 0.2 g of plant leaves were homogenized in 2 mL of K-P buffer (pH 7.6) and centrifuged at 15,000 g for 20 min at 4 °C. Subsequently, the supernatant was removed, and the reactions were prepared for each enzyme activity, as detailed below. The supernatant obtained from the aforementioned extraction was utilized for all enzyme activity readings. As enzyme activities were calculated in relation to protein content, it was first necessary to determine the protein content.

To determine SOD activity, the following solution was prepared: 2.9 mL K-P buffer (pH 7.6), 0.5 mL sodium bicarbonate, 0.5 mL methionine, 0.5 mL nitro blue tetrazolium chloride (NBT), 0.5 mL riboflavin, and 0.1 mL sample supernatant. This solution was then placed in glass test tubes. The tubes were gently mixed and then incubated under light for 10–15 min (until the color turned blue). The absorbance at 560 nm was then determined with a spectrophotometer. The quantity of enzyme was calculated in micromoles per milligram of protein.

For the determination of APX activity, 0.7 mL of K-P buffer (pH 7.6), 0.1 mL of $H_2O_2$, 0.1 mL of the sample supernatant, and 0.1 mL of ascorbic acid were added, and absorbance readings were taken at 290 nm for 1 min in a spectrophotometer. The enzyme amount was calculated as micromoles per milligram of protein per minute.

The CAT activity was determined by adding 0.8 mL K-P buffer (pH 7.6), 0.1 mL sample supernatant, and 0.1 mL $H_2O_2$, and absorbance readings were performed at 240 nm for 1 min. The amount of enzyme that decreased the absorbance by μmol in 1 min at 25 °C was calculated as nmol/mg protein/min.

## HSP70 and HSP90 gene expression levels

The RNeasy Plant Mini Kit (Qiagen, Valencia, CA, USA) was used to isolate total RNA from 100 mg of ground-up leaves that had been frozen in liquid nitrogen. Following the

**Table 1 Primer sequences of genes analyzed for expression.**

| Gene name | Forward primer | Reverse primer |
|---|---|---|
| ZmActin1 (accession: J01238) | ATCACCATTGGGTCAGAAAGG | GTGCTGAGAGAAGCCAAAATAGAG |
| ZmHSP90 (accession: EE257979) | ATCTGGCACTTCAGGAACAGG | AACGCCTCCATTGCTTCGTAT |
| ZmHSP70 (accession: DY307167) | TGCTTGACGTCACTCCTCTC | CTCGTACACCTGGATCAACACA |

isolation of RNA, the purity of the RNA was quantified using a nanodrop device (Thermo ND2000, Waltham, MA, USA). RNA samples with an A260/A280 ratio of approximately 2.0 were utilized for cDNA synthesis. Total RNA was isolated once more from samples exhibiting values below the desired threshold, and this process was repeated until the requisite value was achieved. The measured RNA concentrations were adjusted to 1,000 ng/µL, and cDNA was obtained by reverse transcription polymerase chain reaction (RT-PCR). The application of RQ1 RNase-free DNase (Promega, Madison, WI, USA) eliminated any DNA contamination. For cDNA synthesis, Procomcure Biotech, Thalgau, Austria, provided the VitaScript TM First-strand cDNA Synthesis Kit (PCCSKU1301). The cDNA samples were then adjusted to 100 ng/µL and the expression levels of the gene products of interest were determined by RT-qPCR (real-time PCR). RT-qPCR conditions: 95 °C 5 min, followed by 40 cycles of 95 °C 15 s, 60 °C 30 s, and 72 °C 30 s. Following the synthesis of cDNA from the prepared RNA samples, a RT-qPCR analysis was conducted using specific primers for the selected genes and maize Actin as a housekeeping gene (Table 1). The study was conducted using the CFX Connect Real-Time PCR Detection System (Bio-Rad, Hercules, CA, USA) and the 2X Magic SYBR Kit (Procomcure, Thalgau, Austria). The $2^{-\Delta\Delta Ct}$ technique was employed to calculate the relative expression (*Livak & Schmittgen, 2001*). Three biological and two technical replicates were employed for each analysis.

## Western blotting for HSP70 and HSP90

The purification of plant samples was conducted in accordance with the methodology proposed by *Abraham-Juárez (2019)*. The Bradford technique was utilized to quantify the protein concentration of the samples. Based on the identified protein concentrations, the sample amounts were adjusted. Subsequently, the samples were subjected to electrophoresis on a 12% polyacrylamide gel. Subsequently, a PVDF membrane with an area equivalent to that of the gel was submerged in 10 mL of 100% methanol for 1 min in a square Petri plate. The transfer procedure was conducted using the ThermoFisher Scientific iBlot 2 Dry Blotting System. The gel-to-membrane transfer procedure was conducted using the sandwich-shaped membrane transfer mechanism of the iBlot 2 device at 20 volts for 7 min. A blocking solution was subsequently formulated following the completion of the transfer procedure. Later, the gel image was detached from the transfer paper and immersed in the pre-prepared blocking solution for a period of 5 min. The blocking solution was then transferred to 2.5 ml tubes, with 2 ml in each tube, which was equivalent to the quantity of antibodies employed. This was for the antibody application

procedure. In this investigation, Agrisera UBQ11 (internal control), HSP70, HSP90, and HRP Goat Anti-Rabbit IgG (H + L) were used as secondary antibodies. Then, the iBind flex card was attached to the membrane cylinder, after which 10 milliliters of blocking solution were wetted. The membrane was immersed in the ECL solution for a period of 2.5 h. A chemiluminescence imaging device was utilized in the imaging procedure. Invitrogen SeeBlue Plus2 Prestained Standard was used as a ladder. Prior to imaging, the membrane was incubated in a 1:1 ECL solution for 5 min in the dark. The final band images from the films were converted to digital media by means of a scanner. A densitometric analysis of the bands was conducted using the American Institute of Health's "Image J" tool (http://rsb.info.NIH.gov/nih-image/).

## Statistical analyses

The study data was subjected to an entirely random factorial design in order to examine for variance using IBM SPSS 26. To identify differences in means, the Duncan multiple comparison test was utilized. The data is presented in bar graphs and error bars, which display the mean and standard error of the data. The figures were generated using IBM SPSS 26. Correlations between the examined parameters were analyzed by Pearson correlation analysis according to $p < 0.05$.

# RESULTS

## Osmoregulation parameters

A statistical analysis revealed that maize subspecies, drought treatments, and their interactions were found to be statistically significant at the 1% level in terms of leaf RWC, LOT, PRO, and SPR content (Table S1).

In the absence of stressors, the RWC of maize subspecies exhibited the highest levels in sugar maize, reaching 95.20% (Fig. 1A). The water content of popcorn and dent maize was 77.48% and 72.43%, respectively. In the context of the moderate drought, the water content of dent maize showed the highest level of resilience, despite exposing the lowest water content under normal conditions. The reduction in water content of dent maize was approximately 4%, while popcorn maize displayed a 35% decline and sugar maize displayed a 44% reduction. In response to severe drought conditions, sugar maize had a 70% reduction in water content compared to the control, while popcorn maize demonstrated a 35% decline in water content at the same rate as moderate drought severity. However, dent maize lost the content of water by approximately 15% compared to the control, resulting in a RWC value of 61.82%. The LOT in maize plants under drought stress was found to be parallel to the RWC. In the dent maize, a 33.42% LOT was observed under severe drought stress. However, even under moderate drought stress, turgor loss exceeding 40% was measured in popcorn and sugar maize. Although turgor loss in sugar maize under severe drought remained at the same level as in moderate drought, LOT in sugar maize reached 62.93% (Fig. 1B).

As drought stress increased, PRO accumulation increased in dent and popcorn maize, however decreased in sugar maize (Fig. 1C). Dent maize, which had the lowest PRO content (12.13 mmol g$^{-1}$FW) under control conditions, reached a PRO value of

Peer J

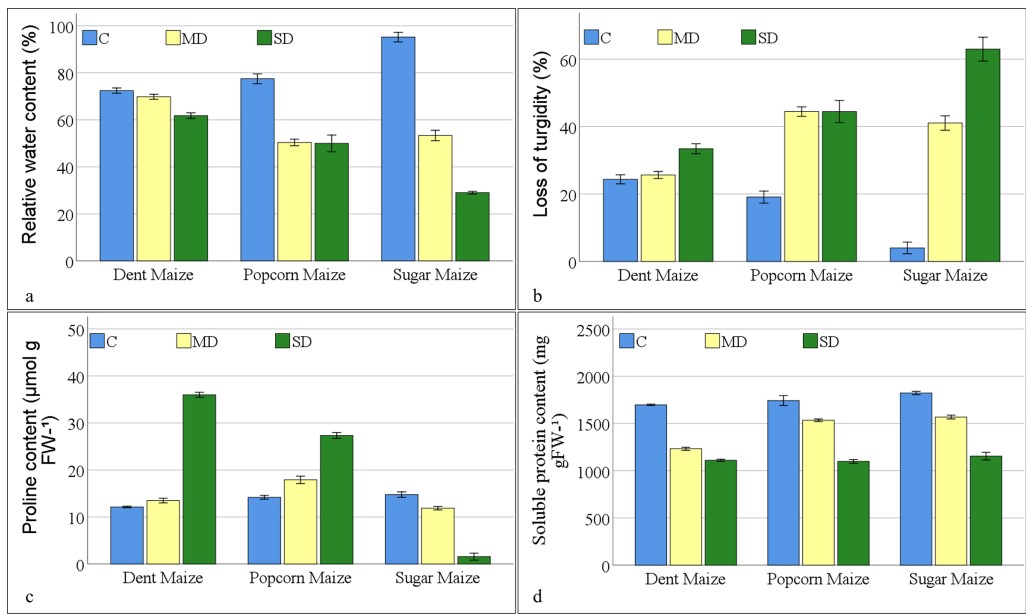

**Figure 1 Changes of osmoregulation parameters for the maize subspecies in drought stress (C: control, MD: moderate drought, SD: severe drought).**

35.99 mmol g$^{-1}$ FW when exposed to severe drought. Popcorn had a higher PRO content than other maize subspecies under both control and moderate drought conditions. In contrast, PRO content reduced in sugar maize as a response to drought stress, reaching a minimum of 1.56 mmol g$^{-1}$ FW in severe drought. The SPR content of maize leaves decreased under drought stress across all maize subspecies. The highest SPR content was observed in sugar maize under all conditions; however, it was the lowest in dent maize. The greatest reduction of SPR in maize leaves belonged to dent maize under moderate drought stress and in sugar maize under severe drought stress in comparison to normal conditions. The highest SPR content was measured in popcorn maize under severe drought stress, in comparison to moderate drought stress (Fig. 1D).

## Membrane injury parameters

The results of variance analyses of membrane injury parameters (Table S1) indicated that there were statistically significant changes in membrane damage (MD%), MDI, MDA, and $H_2O_2$ content among maize subspecies, as well as in response to drought stress. The interaction between maize subspecies and drought stress was found to be insignificant for MD% and MDI. However, significant interactions were observed for MDA and $H_2O_2$ content.

The greatest degree (19.36%) of MD%, as indicated by electrolyte leakage, was observed in sugar maize subjected to severe drought stress. Membrane damage increased with the severity of drought stress in all maize subspecies. Dent maize indicated the least damage (1.20%) under moderate drought stress, while popcorn maize exhibited the least damage (3.60%) under severe drought stress (Fig. 2A). As anticipated, the MDI demonstrated a decline in accordance with the severity of the applied stress. The MDI calculated a range of

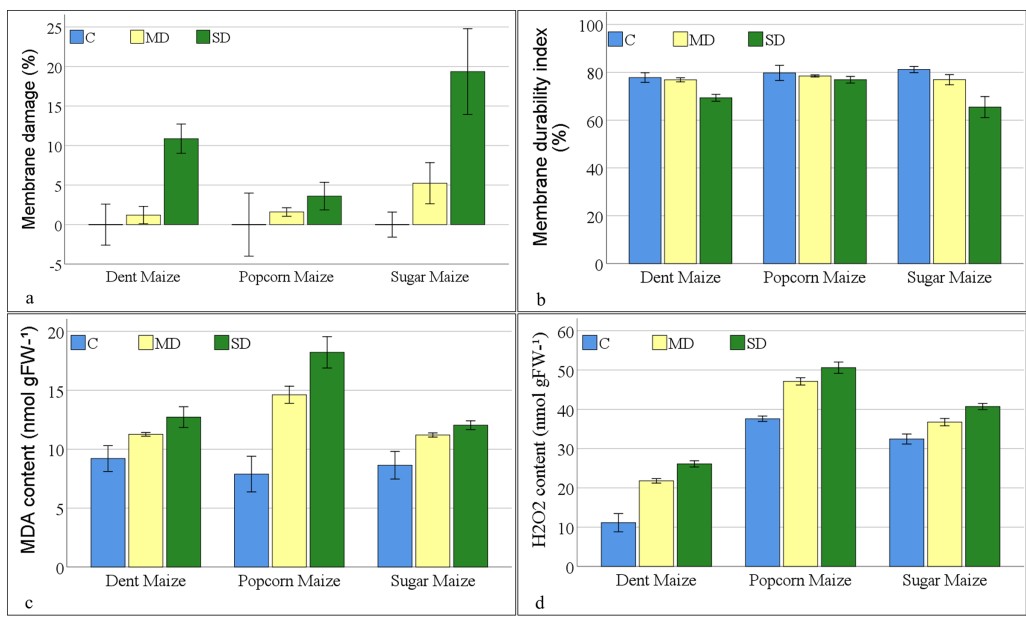

**Figure 2 Changes of membrane injury parameters for the maize subspecies in drought stress.**

values, spanning from 65.44% (sugar maize under severe drought stress) to 81.16% (sugar maize under control conditions). The MDI values were close in popcorn maize in all conditions. Notwithstanding the fact that the highest MDI was observed in sugar maize under control conditions, it was found in popcorn maize under moderate and severe drought (Fig. 2B).

The MDA values of maize plants under control conditions ranged from 7.89 to 9.21 nmol gFW$^{-1}$ (Fig. 2C). Popcorn maize had the lowest MDA values, while dent maize had the highest. When maize plants were exposed to moderate drought stress, the MDA values increased by 22.34% in dent maize, 85.30% in popcorn maize, and 29.73% in sugar maize. As the severity of the drought increased, the MDA content increased by 7.42%, 12.95% and 24.63% to 12.03 nmol gFW$^{-1}$ in sugar maize, 12.72 nmol gFW$^{-1}$ in dent maize, and 18.22 nmol gFW$^{-1}$ in popcorn maize, respectively. The highest increase was in popcorn maize at both stress severities. Since MDA content was high in dent maize under normal conditions, the increase in stress conditions was lower than in the other subspecies. Nevertheless, the MDA content of sugar maize was comparatively lower increase when the drought severity was altered from moderate to severe.

The lowest levels of $H_2O_2$ were observed in dent maize, while the highest levels were in popcorn maize under all conditions (Fig. 2D). Even though the levels were low, the increase in $H_2O_2$ content was more pronounced in dent maize as drought stress intensified compared to other subspecies. In the absence of drought stress, the $H_2O_2$ content of dent maize was 11.14 nmolg$^{-1}$FW, that of popcorn maize was 37.59 nmolg$^{-1}$FW, and that of sugar maize was 32.44 nmolg$^{-1}$FW. When moderate drought stress was imposed, the $H_2O_2$ content of dent maize increased by 10.65 units. This increase is 4.30 units in sugar maize. In the case of severe drought conditions, the $H_2O_2$ content increased by 13 units in
popcorn maize in comparison to normal conditions. In contrast, it increased by only 3.49 units when the drought changed from moderate to severe, representing the lowest increase among the subspecies.

## Antioxidant enzyme activities

A significant change was observed in the antioxidant enzyme activities of maize plants exposed to drought stress. The interaction between maize subspecies and drought stress was not significant for SOD activity (Table S1). The results of the study indicated that all antioxidant enzymes examined exhibited a parallel increase in the severity of drought stress in all maize subspecies (Fig. 3). The greatest increase in SOD enzyme activities was observed in dent maize plants under moderate drought stress. In severe drought conditions, the highest SOD activity was observed in popcorn and sugar maize. Dent maize revealed similar SOD activity under both moderate and severe drought conditions.

The increases in APX activities of maize plants were close between dent and popcorn maize under moderate drought conditions in comparison to control conditions. However, the increase was twice as great in sugar maize. Severe drought resulted in a 2.6-fold increase in APX activity in sugar maize in comparison to control conditions, while it increased 1.8-fold in popcorn maize and 1.4-fold in dent maize (Fig. 3). The activity of CAT increased rapidly in dent and sugar maize under moderate drought conditions, while it increased more slowly in popcorn maize. In the case of severe drought, the highest CAT activity was observed in sugar maize. In severe drought, CAT activity increased by more than fourfold in sugar and popcorn maize compared to the control. In the transition from moderate to severe drought stress, CAT activity increased by a factor of 2.7 in popcorn maize. Although the magnitude of the increase was considerable, CAT activity was the lowest in popcorn maize (Fig. 3).

## HSP70 and HSP90 gene expression levels

The expression of the HSP70 gene differed among maize subspecies, however was not significant for drought stress or the maize subspecies × drought stress interaction. In contrast, the expression of the HSP90 gene was statistically significant for all experimental variants (Table S1).

The expression levels of the HSP70 gene indicated notable differences among maize subspecies under conditions of drought stress (Fig. 4). In dent maize, the expression level increased from 1.00 to 1.52 under moderate drought stress and decreased to 1.31 under severe drought. In popcorn maize, there was a gradual increase in gene expression levels with increasing drought severity, with levels being slightly higher than those observed in other maize subspecies under all conditions. In contrast, the level of HSP70 gene expression in sugar maize showed a decrease with increasing drought severity, reaching a highly low level.

The level of gene expression of HSP90 was significantly elevated and overexpressed in sugar maize under conditions of severe drought. In dent maize, the gene expression level decreased from approximately 1.00 under normal conditions and moderate drought to 0.71 under severe drought. In popcorn maize, the expression of HSP90 increased by a

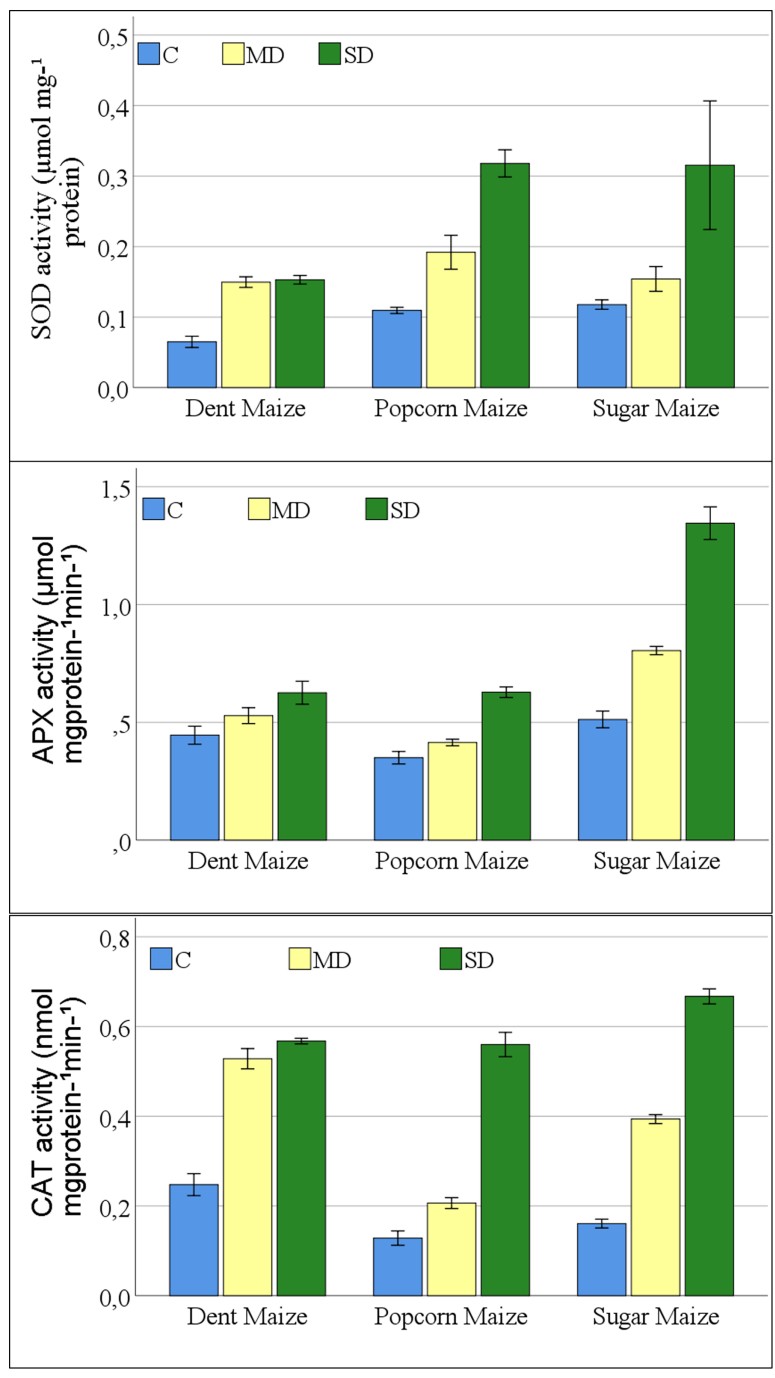

**Figure 3 Changes of antioxidant enzyme activities for the maize subspecies in drought stress.**

factor of 1.5 in the presence of moderate drought in comparison to normal conditions and decreased by a factor of 0.8 in the presence of severe drought. In sugar maize, expression levels demonstrated a marked increase with the progression of drought severity. The HSP90 gene in sugar maize expressed a 3.4-fold increase under moderate drought

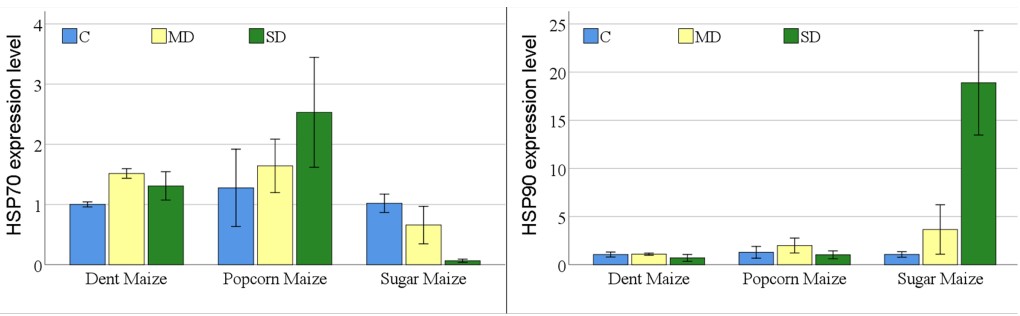

**Figure 4 Changes of HSP70 and HSP90 gene expression levels for the maize subspecies in drought stress.**

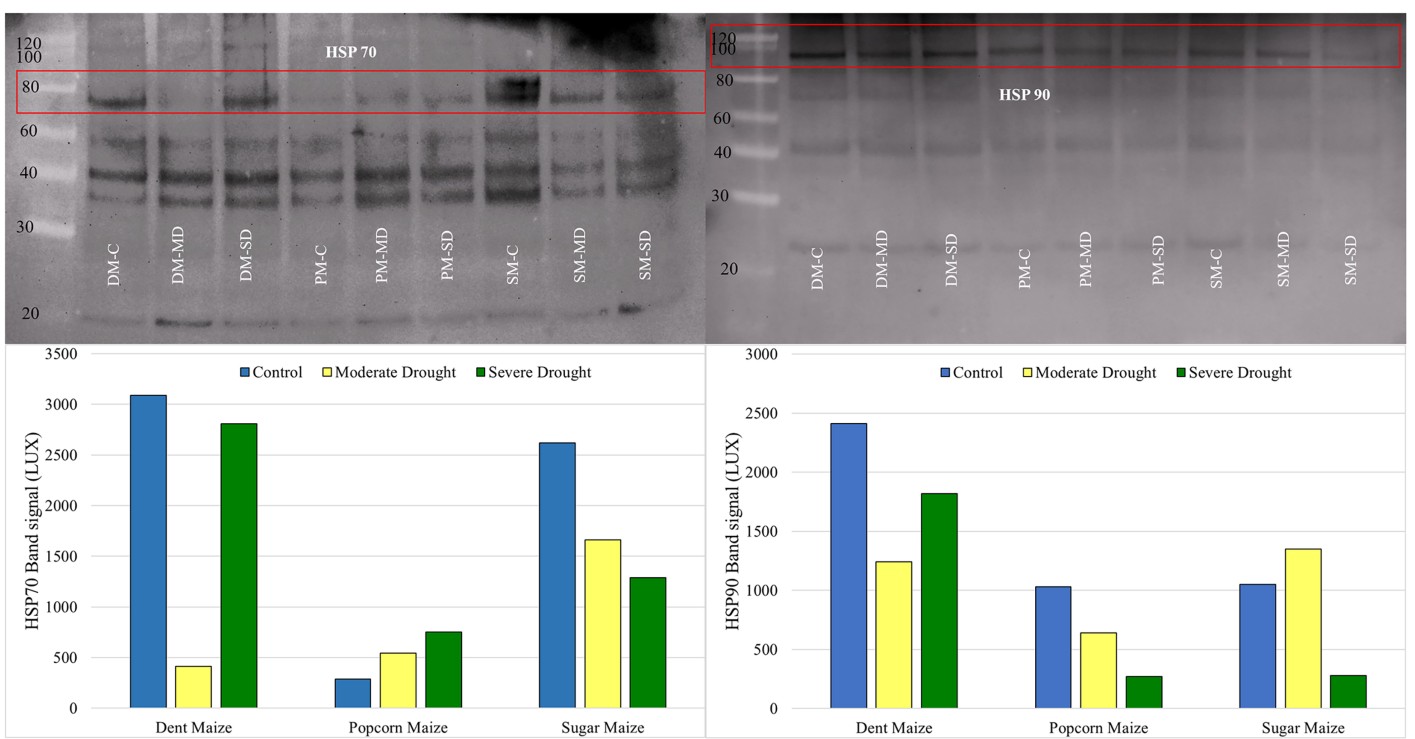

**Figure 5 Western blotting of the HSP70 and HSP90 proteins extracted from maize subspecies in drought stress.**

conditions and a 17.7-fold increase under severe drought conditions relative to normal conditions.

## Western blotting for HSP70 and HSP90

Western blot analysis revealed that the band intensity of the HSP70 protein was highest in dent maize under normal conditions (Fig. 5). The next highest level of HSP70 protein was observed in sugar maize, although this was considerably lower in popcorn. Under the moderate drought, the band intensity of HSP70 was found to be relatively similar in dent and popcorn maize, while exhibiting a higher level in sugar maize. In severe drought, the

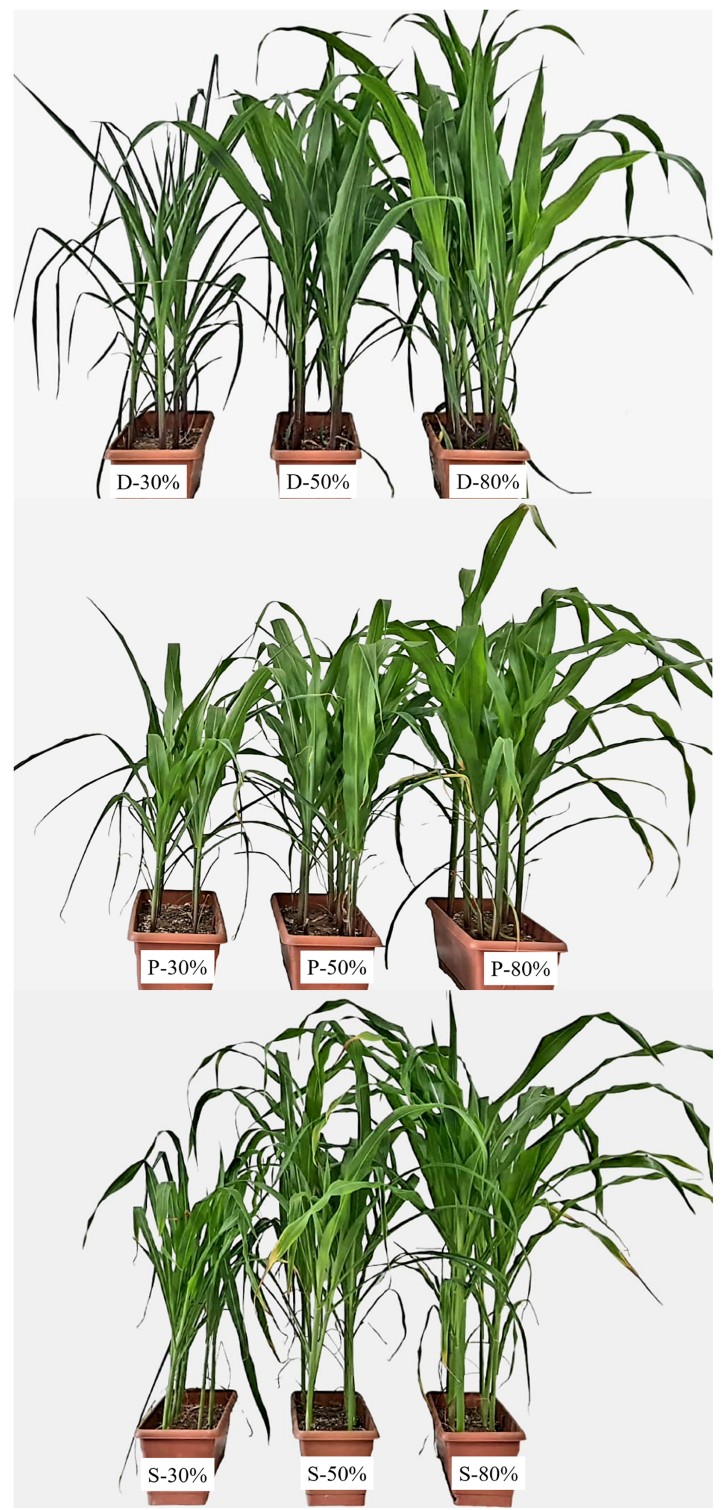

**Figure 6  Morphological appearance of maize plants under drought stress (D: dent maize, P: popcorn maize, S: sugar maize).**
**Table 2  Pearson correlation coefficients between the characteristics analyzed in the study.**

|        | MDI      | RWC      | LOT      | MDA      | PRO    | PRT      | H$_2$O$_2$ | SOD      | APX      | CAT      | HSP70   | HSP90    |
|--------|----------|----------|----------|----------|--------|----------|-----------|----------|----------|----------|---------|----------|
| MD%    | −0.97**  | −0.64**  | 0.64**   | 0.19     | −0.12  | −0.51**  | 0.16      | 0.41*    | 0.74**   | 0.60**   | −0.21   | 0.61**   |
| MDI    |          | 0.61**   | −0.61**  | −0.20    | 0.03   | 0.58**   | −0.02     | −0.35    | −0.65**  | −0.64**  | 0.16    | −0.54**  |
| RWC    |          |          | −0.99**  | −0.56**  | 0.14   | 0.65**   | −0.47*    | −0.68**  | −0.71**  | −0.66**  | 0.08    | −0.61**  |
| LOT    |          |          |          | 0.56**   | −0.14  | −0.63**  | 0.48*     | 0.70**   | 0.69**   | 0.64**   | −0.07   | 0.65**   |
| MDA    |          |          |          |          | 0.42*  | −0.62**  | 0.54**    | 0.61**   | 0.16     | 0.50**   | 0.29    | 0.02     |
| PRO    |          |          |          |          |        | −0.35    | 0.06      | 0.00     | −0.41*   | 0.13     | 0.50**  | −0.56**  |
| PRT    |          |          |          |          |        |          | −0.21     | −0.63**  | −0.50**  | −0.93**  | −0.17   | −0.29    |
| H$_2$O$_2$ |      |          |          |          |        |          |           | 0.60**   | 0.22     | 0.09     | 0.21    | 0.21     |
| SOD    |          |          |          |          |        |          |           |          | 0.58**   | 0.60**   | 0.09    | 0.61**   |
| APX    |          |          |          |          |        |          |           |          |          | 0.70**   | −0.41*  | 0.79**   |
| CAT    |          |          |          |          |        |          |           |          |          |          | −0.07   | 0.44*    |
| HSP70  |          |          |          |          |        |          |           |          |          |          |         | −0.44*   |

**Notes:**
* $p \leq 0.05$.
** $p \leq 0.001$, insignificant ones are not marked.

band intensity of HSP70 was ranked from largest to smallest in dent, sugar, and popcorn maize. The manifestation of changes in HSP70 protein expression under drought stress differed between maize subspecies. HSP70 expression indicated a gradual increase with drought in popcorn maize, however a decline in sugar maize.

The intensity of the HSP90 protein band, which is typically high in dent maize, was found to be equivalent in popcorn and sugar maize and approximately half that observed in dent maize. Moderate drought stress resulted in a reduction in the quantity of HSP90 protein in dent and popcorn maize in comparison to the control conditions, whereas an increase was observed in sugar maize. In response to severe drought stress, HSP90 protein expression in dent maize increased gradually, although it remained below the levels observed under normal conditions. However, a notable decline was observed in popcorn and sugar maize (Fig. 5).

The aforementioned changes are also evident in plant morphology. As illustrated in Fig. 6, there was a notable shortened plant height, decrease in leaf area as the drought intensified, as well as a decline green color intensity due to the decrease in chlorophyll content.

## Correlation analyses for studied parameters

The results of the two-tailed Pearson correlation analysis conducted to determine the relationships between the parameters examined in the study are presented in Table 2. In particular, there was a positive and highly significant correlation between HSP70 gene expression and PRO content, while there was a negative and significant correlation between HSP70 gene expression and APX activity. The expression of the HSP90 gene was found to be positively and significantly correlated with MD%, LOT, SOD, APX, and CAT activities. Conversely, it was negatively and significantly correlated with MDI, RWC, and PRO.

## DISCUSSION

One of the primary abiotic factors influencing agricultural productivity is a scarcity of water. Drought stress, which is the consequence of water scarcity, can give rise to a multitude of factors that can be detrimental to plants. However, plants have evolved a multitude of physiological, biochemical, and molecular adaptation mechanisms in response to the intricate process of tolerance to drought stress (*Ren et al., 2022*; *Schmitt, do Amaral Junior & Kamphorst, 2024*). Water loss due to drought stress is manifested by a decrease in RWC in plants. It was observed that the RWC values of the maize genotypes studied in this research decreased when exposed to drought (Fig. 1A). The drought-induced reduction in RWC of dent maize, which already has a low RWC under normal conditions, was significantly lower than that of popcorn and sugar maize. This suggests that dent maize is better at maintaining leaf water coverage during water scarcity. Popcorn maize had similar water content in both moderate and severe droughts compared to normal conditions. On the other hand, sugar maize lost a significant amount of water as the drought intensified. *Jain et al. (2019)* suggested that the maintenance of high RWC represents a mechanism for drought resistance rather than drought escape and that it results from adaptive traits such as osmotic adaptation. Under drought conditions, the classical control system involves stomatal closure as a consequence of guard cell turgor. Differences in the stomatal response to water stress help determine the relative ability of species to cope with drought conditions. The decrease in stomatal conductance can be attributed to the decrease in RWC resulting in the LOT (*Badr & Brüggemann, 2020*). The decrease in RWC also indicates a decrease in plant turgor, and in our study, Fig. 1B shows that maize genotypes lost turgor in parallel with the decrease in RWC.

In water-stressed plants, compatible solutes act as osmoprotectants and mediate osmotic correction (*Mukarram et al., 2021*). Of these, free proline is the most abundant osmolyte found in plants grown under water deficit conditions. As a result, high levels of proline can improve water-holding capacity (*Frimpong et al., 2021*). In this study, water deficit increased PRO content in the leaves of dent and popcorn maize (Fig. 1C). The RWC and turgor capacity of these subspecies were also high. This information is supported by the fact that the PRO content of sugar maize, whose water and turgor loss increased with drought stress, decreased with drought stress. *Abdul Mohsin & Farhood (2023)* reported that proline helps plants maintain water cover by acting as an adhesive water shell that shows strong resistance to changes caused by water stress conditions. Drought stress can cause both qualitative and quantitative changes in plant proteins. Plants grown under environmental changes are subjected to changes in protein content, which in many cases leads to a decrease in protein content due to an increase in proline content. Basically, proline accumulation has been reported as a result of stress-induced protein hydrolysis or oxidative inhibition of protein synthesis in plants (*Mansour & Salama, 2020*). In this study, drought stress caused a decrease in SPR content in all maize genotypes. The decrease in SPR content was less in dent maize, where PRO accumulation was high (Fig. 1D). Under abiotic stress conditions, plants that survive the damaging effects tend to increase osmotic

potential at the cellular level by accumulating solutes such as proline (*Ozturk et al., 2021*). However, it has been reported that under drought stress, the total amount of soluble proteins decreases while the levels of proline and antioxidant enzymes increase (*Kosar et al., 2020*).

The most straightforward and quantifiable indicator of the impact of drought on plants is the MDI, which is calculated based on electrolyte leakage (*Rudolphi-Szydło et al., 2022*). A high value of this index indicates that electrolyte leakage in plants is low, membrane damage does not occur, and genotypes with high MDI may indicate high drought tolerance (*Guizani et al., 2023*; *Yin et al., 2024*). In this study, it was observed that the MD% of maize genotypes was higher under severe drought conditions than in the control (Fig. 2A). However, the MDI values were found to be close to the control in all maize subspecies under moderate drought conditions (Fig. 2B). In severe drought conditions, only sugar maize exhibited a reduction in MDI values that was comparable to the control. However, it still did not reach a very low value compared to other genotypes. The greater reduction in MDI of sugar maize under severe drought was due to the it's higher values under control conditions.

Another factor that contributes to a reduction in membrane integrity and cellular structure when plants are subjected to drought is the presence of ROS, which results in an increase in lipid peroxidation (*Ru et al., 2023*). The levels of lipid peroxidation (MDA content) were approximately 8–9 nmol $g^{-1}$ FW at control in all maize genotypes (Fig. 2C), however following the severe drought treatment, they rose to above 18 nmol $g^{-1}$ FW in popcorn maize and above 12 nmol $g^{-1}$ FW in dent and sugar maize. The MDA content of popcorn was found to be higher than that of others in both drought-stress conditions.

Despite its involvement in numerous signaling pathways, $H_2O_2$ becomes toxic and causes cellular damage when it is not removed to maintain its concentration below the threshold for toxicity (*Wang et al., 2024*). The concentration of $H_2O_2$ was found to increase in all maize genotypes under conditions of drought stress (Fig. 2D). Higher levels of $H_2O_2$ were observed in popcorn. Although the lowest $H_2O_2$ content was observed in dent maize under all conditions, a low level of $H_2O_2$ accumulation was apparent in sugar maize when it transitioned from normal to drought conditions. As the concentration of $H_2O_2$ increases, it reacts with lipids and proteins, resulting in the oxidation of lipids and damage to the cell membrane. As with MDA content, it indicates the level of lipid peroxidation and membrane damage. However, MDA content is more positively related to water loss under drought conditions (*Singh et al., 2022*).

In this study, antioxidant enzyme activities (SOD, APX, and CAT) increased as a response to increasing MDA and $H_2O_2$ content (Fig. 3). All antioxidant enzyme activities examined in the study showed the highest increase in sugar maize. However, its SOD and APX activities were also high under normal conditions. All antioxidant enzymes reached the highest level in all maize genotypes under severe drought stress. The antioxidant system needs to be activated immediately upon the onset of drought stress. These antioxidant enzymes protect plants from oxidative damage caused by ROS and maintain the balance between ROS production and scavenging under stressful conditions (*Rajput et al., 2021*). Furthermore, there is a strong correlation between the level of oxidative

damage caused by ROS in cells and the activity of antioxidants. However, the resistance or susceptibility of the genotype to drought stress is determined by the correlation between antioxidants and ROS generation (*Anjum et al., 2016*).

Drought stress has been found to affect the normal amount and quality of plant protein (Fig. 1D). Drought-induced dehydration triggers stress-related proteins, including HSPs. The primary regulators of plant stress responses are genes encoding HSPs, just like antioxidants. HSPs have been characterized as proteins whose concentration increases significantly at higher temperatures for plant growth; however, recent reports suggest that HSPs are proteins that help fold newly formed proteins or stop misfolding proteins (*Khan & Shahwar, 2020*). *Li et al. (2021a)* reported that most of the HSPs in maize, including the HSP70, the HSP90, and the small HSP family, increased under severe drought stress. In this study, HSP70 and HSP90, whose importance is emphasized in many stress conditions, were determined at the transcriptional level by real-time PCR and at the translational level by Western blotting. The expression of the HSP70 gene was found to be upregulated by drought stress in dent and popcorn maize, while it was downregulated in sugar maize. With regard to the HSP90 gene, there was not much change in its expression level with drought stress in dent and popcorn maize. However, it has been observed to be overexpressed in sugar maize, especially in severe droughts (Fig. 4). The results of the WB analysis produced disparate outcomes when compared to those of the real-time PCR analysis in dent maize (Fig. 5). The expression of HSP70 and HSP90 was found to be higher in control conditions than in drought conditions. HSP70 and HSP90 have also been demonstrated to play a pivotal role in the normal development of plants. These proteins are present at significant levels in normal, unstressed cells, maintained in the cytoplasm at specific stages of the cell cycle, or during development in the absence of stress (*Li & Howell, 2021*). The high expression of HSP70 and HSP90 at the translational level in the absence of stress can be attributed to the aforementioned information.

The expression of HSP70 in popcorn and sugar maize was found to be similar in both assays, with the results supporting each other. HSP70 was upregulated in popcorn in response to stress, whereas in sugar maize, it was downregulated. HSP70s are classified as chaperones and fulfill a number of vital functions. These include promoting the functioning of the antioxidant system, eliminating excess ROS or damaged proteins under stress, and assisting newly generated proteins in folding correctly (*Ul Haq et al., 2019*). In previous studies, the upregulation of HSP70 has been associated with drought tolerance in rice, Arabidopsis, and maize (*Hu et al., 2010*; *Pulido, Llamas & Rodriguez-Concepcion, 2017*; *Devarajan et al., 2021*; *Li et al., 2021b*). Similar to the findings for HSP70 in dent maize, which demonstrated a reduction in expression levels in response to drought stress, the expression of HSP90 was found to be upregulated in severe drought conditions relative to moderate drought. In contrast to popcorn maize, which exhibited a downregulation in drought stress, sugar maize displayed an upregulation in moderate drought and a downregulation in severe drought. Previous studies have demonstrated a positive correlation between the HSP90 gene and drought tolerance in maize plants (*Li & Howell, 2021*; *Li et al., 2021a*, *2021b*; *Liu et al., 2021*). It is well established that the expression of HSP genes is subject to control at both the transcriptional (mRNA) and translational

(protein) levels. Consequently, it is important to recognize that the disparate outcomes observed in the two methods may be attributable to post-transcriptional regulatory mechanisms.

The physiological and molecular mechanisms that occur as a consequence of drought stress in the maize plant serve to determine its degree of exposure to and tolerance to stress, and these are interrelated. The positive correlation between MD%, LOT, and MDA content with antioxidant enzyme activities and HSP90 gene expression (Table 2) indicates that the plant's defense mechanism is activated in response to the negative impacts of stress. It has been demonstrated that an increase in lipid peroxidation can also alter a cell's transcriptional profile, despite the conventional understanding that an elevation in MDA levels is indicative of stress-related damage (Zhang et al., 2024). Moreover, recent research suggests that MDA may function as a stress-signaling molecule in plants, activating genes associated with heat shock/dehydration and antioxidant machinery (Morales & Munné-Bosch, 2019). The expression of the HSP70 gene was found to be correlated with the PRO content and the activity of the APX enzyme. In contrast, the expression of the HSP90 gene was correlated with all properties except for the MDA, $H_2O_2$, and SPR content. It is also noteworthy that a negative correlation was identified between the expression of the HSP90 gene and the HSP70 gene in this study. Every HSP class comprises members with distinct roles. Nevertheless, a fundamental component of the integrated HSP machinery is collaboration throughout different HSP networks (Wang et al., 2004). In response to stressful circumstances, HSPs utilize ROS as signaling molecules, thereby preventing the aggregation of proteins. Earlier research has indicated intricate relationships between HSPs and plant responses to ROS (Ding, Li & Zhang, 2022; Singh et al., 2019). In order to clarify stress tolerance in maize, Li & Howell (2021) proposed that future studies should examine the interactions between HSPs and other stress response systems. The results of this research may provide new answers for the interactions that are still under investigation.

The traits analyzed in this study are not independent of each other; rather, their combined and cumulative effects are responsible for the drought tolerance of the plant. Ultimately, this also affects the morphological characteristics and yield of the plant, providing precise information about its drought tolerance. In our previous study, when we evaluated the morphological characteristics and chlorophyll content of maize plants, we found that they decreased the most in sugar maize (Eskikoy & Kutlu, 2024). It can be concluded that dent corn is the most drought-tolerant subspecies and sugar maize is the most sensitive. This study has demonstrated the importance of HSPs in determining drought tolerance, with HSP90 being more effective in selecting sensitive genotypes.

## CONCLUSIONS

Drought stress represents one of the most deleterious environmental disturbances in maize cultivation. This study has revealed that moderate to severe drought stress alters physio-biochemical, enzymatic, and molecular traits in dent, popcorn, and sugar maize subspecies. The higher tolerance of dent maize to drought stress is associated with its conserved water content, high proline accumulation, increased antioxidant activities, and lower lipid

peroxidation compared to other subspecies. The high membrane damage, water and turgor loss, increased enzyme activities, and HSP90 gene expression in response to increases in MDA and $H_2O_2$ content in sugar maize, especially under severe drought, indicate that it is heavily affected by stress. Popcorn maize showed a tolerance in the middle of the other two subspecies. The results revealed that different physiological, biochemical, enzymatic and molecular responses of maize subspecies were dependent on the severity of drought. Furthermore, our findings indicate that distinct transcriptional and translational systems exist between moderate and severe drought stress.

## ACKNOWLEDGEMENTS

This manuscript is based on Gokhan Eskikoy's master's thesis, which was supervised by Imren Kutlu.

### Funding
This research was supported by the Scientific Research Projects Commission of Eskisehir Osmangazi University, grant number FYL-2023-2791. The funders had no role in study design, data collection and analysis, decision to publish, or preparation of the manuscript.

### Grant Disclosures
The following grant information was disclosed by the authors:
Eskisehir Osmangazi University: FYL-2023-2791.

### Competing Interests
Imren Kutlu is an Academic Editor for PeerJ.

### Author Contributions
- Gokhan Eskikoy performed the experiments, analyzed the data, prepared figures and/or tables, authored or reviewed drafts of the article, and approved the final draft.
- Imren Kutlu conceived and designed the experiments, performed the experiments, analyzed the data, prepared figures and/or tables, authored or reviewed drafts of the article, and approved the final draft.

### Data Availability
Raw data are available in the Supplemental Files.

### Supplemental Information
Supplemental information for this article can be found online at http://dx.doi.org/10.7717/peerj.17931#supplemental-information.

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
