# Peer review of "Inter-subspecies diversity of maize to drought stress with physio-biochemical, enzymatic and molecular responses"

_PeerJ, doi:10.7717/peerj.17931_

## Round 0.1 · original submission · Minor Revisions

Dear Authors
The manuscript needs a minor revision before being reconsidered for publication. The authors are invited to revise the paper considering all the suggestions made by the reviewers. Please note that the requested changes are required for publication.
With Thanks

Reviewer 1 ·

Basic reporting

Three different drought levels and three subspecies of maize were used in the study. Nine physico-biochemical, enzymatic and molecular properties were examined. Clear and unambiguous professional English was used in the study.Literature references and sufficient field history/context are provided in the research.

Experimental design

The employee's experimental setup was well planned. The methods are explained with sufficient detail and information to be repeated in the study.The traits discussed in the research were obtained with appropriate methods and analyzed with an suitable statistical design.

Validity of the findings

In the study, the findings were discussed comparatively with the literature. And the conclusion is written in a way that highlights important issues.

Additional comments

I recommend using the abbreviation names of the traits to ensure unity in all tables and figures.

Reviewer 2 ·

Basic reporting

The manuscript is well written and interesting

Experimental design

The study demonstrates a good experimental design

Validity of the findings

No comments

Additional comments

This research investigated the physiological and molecular responses of three maize subspecies (dent, popcorn, and sugar) to drought stress. The study analyzed various parameters including relative water content (RWC), leaf osmotic potential (LOT), proline (PRO) content, soluble sugar content (SPR), membrane damage (MD%), membrane integrity index (MDI), malondialdehyde (MDA) content, hydrogen peroxide (H2O2) content, and the expression of heat shock protein genes (HSP70 and HSP90). The findings are interesting and can contribute to breeding programs for developing drought-resistant maize varieties.
-Comments and Suggestions for Authors
-The abstract could benefit from a more concise yet informative summary of key results. For instance, instead of listing all analyzed parameters, mention the most significant findings that differentiate dent maize's tolerance mechanisms.
-In abstract, consider including a specific example of a measured parameter (e.g., higher RWC in dent maize under severe drought) to strengthen the impact.
- The introduction provides a clear background on maize and its economic importance and effectively highlights the threat of drought stress to maize production. However, it can be shortened by condensing redundant information. For instance, some details about specific subspecies uses (dent, popcorn, sugar) can be omitted.
- While the introduction covers many aspects of drought stress response, consider emphasizing the specific focus of the research: comparing drought tolerance mechanisms between maize subspecies and the role of HSPs.
- Briefly mention the chosen maize subspecies (dent, popcorn, sugar) earlier in the introduction for better context.
-The section of Materials & Methods provides a comprehensive overview of the experimental design and analysis methods. It allows for a good understanding of how the researchers investigated drought tolerance mechanisms in different maize subspecies. However, it is not clear for me what is the specific growth stage at which drought stress was applied.
- In results section, the study mentions morphological changes observed under drought stress (Figure 6). Including a description of these changes would strengthen the overall analysis.

Reviewer 3 ·

Basic reporting

Abstract
Background. The background section emphasizes the importance of the subject. In addition, the important issue of today's important varieties of corn, such as dent corn, popcorn and sweet corn, is drought and its effect mechanisms on plants have been studied.


Methods.
Three distinct irrigation regimes were employed to assess the impact of varying levels of drought stress on maize plants at the V14 growth stage; 80% field capacity, 50% field capacity and severe drought (30% field). Leaf relative water content (RWC), loss of turgidity (LOT), proline (PRO) and soluble protein (SPR) contents, membrane durability index (MDI), malondialdehyde (MDA), and hydrogen peroxide (H2O2) content, the antioxidant enzyme activities of superoxide dismutase (SOD), ascorbate peroxidase (APX), and catalase (CAT). was analyzed by growing all plants under controlled conditions.



Results.
In the results section, the results of all three corn varieties are given in full. For example (line 37-39), dent maize, which is capable of maintaining its RWC and turgor in both moderate and severe drought, and employs its defense mechanism effectively by maintaining antioxidant enzyme activities at a certain level despite less

Introduction.
In this section, it is mentioned that the corn plant is an important cereal plant and its use in humans, animals and industry. While making these explanations, quotes were made from many researchers. For example, Popcorn maize (Zea mays everta) and sugar maize (Zea mays saccharata), on the other hand, are commonly consumed as snacks and in human nutrition (Pandita et al., 2023). Besides maize has always been of great importance due to its use as human food, animal feed, and in numerous industrial products and is cultivating large areas, climate change cause abiotic issues such as drought stress, which poses a significant risk to maize production (Kim and Lee, 2023).
31 articles were cited on the subject. This is a positive approach in terms of examining the issue better. Some of them: Adewale et al., 2018; Rasheed et al., 2023. Seleiman et al., 2021; Sanders and Arndt, 2012, Sapes and Sala, 2021; Lambers et al., 2019.
Relevant quotes are given in the "references" section.

Experimental design

Methods.
Three distinct irrigation regimes were employed to assess the impact of varying levels of drought stress on maize plants at the V14 growth stage; 80% field capacity, 50% field capacity and severe drought (30% field). Leaf relative water content (RWC), loss of turgidity (LOT), proline (PRO) and soluble protein (SPR) contents, membrane durability index (MDI), malondialdehyde (MDA), and hydrogen peroxide (H2O2) content, the antioxidant enzyme activities of superoxide dismutase (SOD), ascorbate peroxidase (APX), and catalase (CAT). was analyzed by growing all plants under controlled conditions.

Validity of the findings

Discussion.
The first sentence in the discussion section of the researchers' study,; It is important to emphasize the work of researchers named Ren et al., 2022 and Schmitt et al., 2024 and cite their own work. On the other hand, it is also positive that it shows the RWC values obtained in its own studies in a figure. Again, the study revealed that one of the main abiotic factors affecting agricultural productivity is water scarcity.
In the research, all values of all three corn varieties were determined and their advantages and disadvantages compared to each other were explained with previous studies. For example; It can be concluded that dent corn being the most
drought-tolerant subspecies and sugar maize the most sensitive. This study has demonstrated the importance of HSPs in determining drought tolerance, with HSP90 being more effective in selecting sensitive genotypes.
Although we have not conducted such comprehensive studies, I have seen in my studies that especially sweet corn is sensitive to water stress conditions. It demonstrates the validity of the research conducted in this study.
Figures and graphs appropriately describe the results

Additional comments

General Explanations
Corn plant is one of the three most important plants grown in the world, along with wheat and rice
Today, climate changes and drought have become very prominent issues
The English grammatical structure used in the article is appropriate. The expressions used are clear and concise. Therefore, it complies with PeerJ standards.
I find it appropriate to publish the article

---

## Round 0.2 · accepted · Accept

Dear Authors,

I have evaluated your revision and I am pleased to inform you that the manuscript has improved after the last revision and can be accepted for publication.

Congratulations on accepting your manuscript, and thank you for your interest in submitting your work to PeerJ.

With Thanks